# Knowledge, attitude, and uptake of human papilloma virus vaccine and associated factors among female preparatory school students in Bahir Dar City, Amhara Region, Ethiopia

**Etenesh Adela Lakneh**[1]*, **Eleni Admassu Mersha**[2], **Melash Belachew Asresie**[2], **Habtamu Gebrehana Belay**[3]

1 Department of Midwifery, Debre Tabor Health Sciences College, Debre Tabor, Ethiopia, 2 School of Public Health, College of Medicine and Health Sciences, Bahir Dar University, Bahir Dar, Ethiopia, 3 Department of Midwifery, College of Health Sciences, Debre Tabor University, Debre Tabor, Ethiopia

* eteneshadela@gmail.com

**Data Availability Statement:** HPV.

**Funding:** The authors received no specific funding for this work.

## Abstract

### Background

The human papillomavirus vaccine is one of the main preventative measures for cervical cancer. However, global vaccine uptake is low; the problem is particularly acute in low and middle-income countries. The purpose of this study is to assess female preparatory school students' knowledge, attitude, and uptake of the human papillomavirus vaccine and associated factors in Bahir Dar City, Ethiopia.

### Methods

Institutional-based cross-sectional study was conducted among 633 female preparatory school students in Bahir Dar city from March 1–30, 2021. Participants were selected using the multistage sampling technique. Data were collected using a structured self-administered questionnaire and entered into Epi-data and exported to SPSS for analysis. Binary and multivariable logistic regression analyses were done using an odds ratio with a 95% confidence interval. Finally-value < 0.05 was considered significant in multivariable analysis.

### Result

The proportion of Human Papillomavirus (HPV) vaccine uptake, knowledge of the vaccine, and respondents' attitudes toward the vaccine were 45.3% (95% CI = 41.6–49.4%), 58.1% (95% CI = 54.4–61.9%), and 16% (95% CI = 13.2–19.5%), respectively. Having a history of sexual contact AOR = 2.80 (95% CI = 1.64–4.76), hearing about HPV infection AOR = 1.59 (95% CI = 1.13–2.24), and having a positive attitude toward HPV vaccine AOR = 1.46 (95% CI = 1.03–2.08) were significantly associated with knowledge about the HPV vaccine. Discussion of reproductive health issues with family AOR = 2.558 (95%CI = 1.800–3.636), and having good knowledge about HPV vaccine AOR = 3.571(95%CI = 2.494–5.113) were associated with a positive attitude toward the HPV vaccine. Good knowledge AOR = 2.36

**Competing interests:** The authors have declared that no competing interests exist.

**Abbreviations:** ACIP, Advisory Committee on Immunization Practices; AOR, Adjusted Odd Ratio; CI, Confidence Interval; COR, Crude Odd Ratio; HPV, Human Papilloma Virus; OR, Odds Ratio.

(95%CI = 1.48–3.76) and a positive attitude toward HPV vaccine AOR = 2.87(95%CI = 1.70–4.85) were strongly associated with HPV vaccine utilization.

## Conclusion

In this study, there was a very low uptake of HPV vaccination among female students, and only a small proportion of them had good knowledge of the HPV vaccine and a favorable attitude toward the HPV vaccine.

## Introduction

Cervical cancer is one of the world's major public health issues, as well as the most common sexually transmitted infection among women. Over 99% of cervical cancer cases are associated with genital infection with certain human papillomaviruses [1]. Cervical cancer is the second most common female malignant tumours worldwide. The necessary cause of cervical cancer has been identified as persistent infection with high-risk human papillomavirus (HPV) [2]. Cancer is estimated to account for approximately 5.8% of total national mortality in Ethiopia [3]. The most common cancers in Ethiopia's adult population are breast cancer (30.2%), cervical cancer (13.4%), and colorectal cancer (5.7%) [4].

Human papillomavirus (HPV) is the most common sexually transmitted infection (STI). Approximately 75% of sexually active people are infected with HPV during their lifetime [5]. More than 100 HPV genotypes are known. Of these, at least 13 genotypes are the cause of cervical cancer and genital and pharyngeal cancers. HPV types (16 and 18) are known to cause 70% of cervical cancers. HPV types 16 and 18 can also cause cancer in other parts of the body, including the vulva, vagina, penis, and anus and 90% of cases of genital warts are caused by HPV types 6 and 11 [6, 7]. The vast majority of HPV infections are asymptomatic or sub-clinical, which has contributed to the rapid transmission and spread of the virus [8].

The human papillomavirus vaccine is one of the most effective ways to prevent HPV infections in women who have never been infected with HPV [9]. The HPV vaccine was first developed by the University of Queensland in Australia and the final form was made by Georgetown University medical center, the University of Rochester, and the United States (U. S). national cancer institute [10]. Food and drug agencies approved the first preventive HPV vaccine under the trade name Gardasil, thereby 64 countries nationally implemented HPV immunization programs [11].

One of the primary prevention strategies for HPV infection is vaccination. HPV vaccination can prevent more than 90% of these cancers [8]. Two HPV vaccines (Gardasil and Cervarix) protect against the two strains of HPV types 16 and 18 [12]. The quadrivalent vaccine is also highly efficacious in preventing genital warts, which are caused by infection with HPV type 6 causes anogenital warts and HPV type 11 causes oropharyngeal cancer [13]. Gardasil works by stimulating the immune system to attack HPV types 6, 11, 16, and 18. Once Gardasil is administered, the body's immune system recognizes the viral proteins in Gardasil as foreign, and develops antibodies against them, thus providing immunity from future infections [14].

The vulnerability of young, sexually active women is recognized and steps are planned in many countries to implement HPV vaccination [15]. The world health organization recommends that the primary target group for HPV vaccination is girls aged 9–14 years and secondary target populations are females aged $\geq$ 15 years [16].

The HPV vaccine can prevent infection and cancer at an early age before any exposure to the virus. The HPV vaccine is recommended for everyone through the age of 26 years. Vaccination usually begins at 11 or 12 years of age, but can be given as early as 9 years of age. The decision to vaccinate should be individualized and discussed with the health care provider for older than 26 years of age [17]. The Advisory Committee on Immunization Practices now recommends a 2-dose series starting at 11 to 12 years [18]. Vaccinating girls before initiation of sexual activity is an important primary prevention intervention in a comprehensive cervical cancer prevention and control program. Exposure to HPV during adolescence and young adulthood may cause cervical cancer in later years. Because of this, Ethiopia had initially planned to introduce the HPV vaccine through a routine immunization program [19].

Previous research concentrated on cervical cancer screening; however, Ethiopia began using the HPV vaccine school-based method in December 2018. Studies on HPV vaccine awareness, attitudes, and uptake were scarce. As a result, the purpose of this study is to investigate HPV vaccine knowledge, attitude, and uptake and also its associated factors among female students in Bahir Dar, Ethiopia.

## Methods and materials

### Study area and period

An institutional-based cross-sectional study was conducted in Bahir Dar City among female preparatory school students in the age group of 15–24 years from March 1–30, 2021. Bahir Dar is the capital city of the Amhara region, which is located 565 km away from Addis Ababa, the capital city of Ethiopia. The estimated total population of Bahir Dar City in 2019 was 445,084. Of this population, 222,097 (49.9%) are males and 222,987 (50.1%) are also females. Bahir Dar city had six government and five private preparatory schools.

### Population

All female students enrolled in government and private preparatory schools for the 2021 academic year in Bahir Dar city were the source population and those who were selected in those schools during the data collection period were the study population.

### Inclusion and exclusion criteria

Female students registered in selected governmental and private preparatory schools in Bahir Dar city were included, while female students who had lived in the city for less than six months and those who were absent during the data collecting period after two separate visits were excluded.

### Sample size determination

Considering the absence of previous data on specific study populations, knowledge, attitude, and uptake of HPV vaccination, the maximum sample size of 50% of prevalence was used in the calculation. The following assumptions were made: by considering the proportion of 50%, 95% confidence interval, and 5% margin of error. Using a design effect of 1.5 and a non-response rate of 10%, the total sample size was 633.

### Sampling procedure

The multi-stage sampling technique was used to recruit participants for the study. First, preparatory schools were classified into public and private institutions, and then, using a simple random sampling process, three out of six governmental institutions and two out of four private

schools were selected. The resulting sample size was then distributed proportionally to the selected schools. The rosters or name lists of students were utilized as a sampling frame to pick study subjects from each grade level. All students were sorted alphabetically. Finally, 633 students were selected using random computer generator software.

### Operational definitions

**Uptake of HPV vaccine.**   Refers to participants who had received at least 1 dose of HPV vaccine.

**Good knowledge about the HPV vaccine.**   When participants answered above or equal to the average number of 10 HPV vaccine knowledge questions, they were considered to have good knowledge of the vaccine.

**Attitudes toward the HPV vaccine.**   When participants scored above or equal to the average score of 12 attitude questions, they were considered to have a positive attitude toward the HPV vaccine.

### Data collection procedure and techniques

The data was collected by using pretested and structured self-administered questionnaires after reviewing previous literature. The questionnaire was first developed in English and translated to the local language Amharic. Three diploma nurses who are familiar with the local language were recruited as data collectors. Two-degree midwives were assigned as supervisors for the data collectors, and the investigators provided overall supervision.

### Data quality control

The training was given to data collectors and supervisors for two days on data collection procedures, the content of the questionnaire, interview techniques, and confidentiality of the information obtained from the respondents regarding the knowledge, attitude, and uptake of HPV vaccine questions. Before conducting the main study, a pretest was carried out on 32(5%) of the sample in school that were not selected for the actual data collection. The principal investigator and supervisor conducted day-to-day on-site supervision during the whole period of data collection. At the end of each day, the questionnaires were reviewed and checked for completeness and accuracy by the supervisor and investigator. Data quality was ensured during collection, entry, and analysis.

### Data processing and analysis

Data were cleaned, coded, and entered by using Epi-Data version 4.6 and then exported to SPSS 21 for analysis. Descriptive statistics were carried out and frequency, tables, and graphs were used to present the descriptive results. On bivariable analysis, all independent variables associated with the dependent variable with $p \leq 0.25$ were entered into multivariable analysis, then the significant association was identified based on $p < 0.05$ with 95% CI. Model fitness was checked by using Hosmer and Lemeshow goodness of fit.

### Ethical approval and consent to participate

Ethical clearance was obtained from the Institutional Review Board of Bahir Dar University College of Medicine and Health Sciences and a supporting letter was obtained from the Bahir Dar town education office. Permission was obtained from each respective Bahir Dar city preparatory school to conduct the study. Furthermore, after providing appropriate information about the study and the right to refuse participation, each respondent provided verbal

informed consent and assent. The parents and guardians of the minors verbally consented before the adolescent females were interviewed. Each school head ensured that participants gave their verbal approval. Respondents were informed that all the data obtained from them was kept confidential and anonymous. To ensure confidentiality, the names of respondents were replaced by code numbers.

## Result

### Socio-demographic characteristics

A total of 633 students were included in the study, with approximately 620 of them correctly answering the questions, yielding a response rate of 98%. The mean age of study participants was 17 years (SD ±1.18). Almost 90% of participants were in the age range of 15–19 years. Three hundred ninety-eight (64.2%) of the participants were in grade 12 and more than two third of participants 482 (67.7%) were orthodox religious followers (Table 1).

### Source of information

Out of a total of 620 participants, 257(41.5%) heard HPV vaccine on radio/television, 178 (29%) heard from teachers, 128(20.6%) heard from the internet and 126(20%) heard from parents (Fig 1).

### Knowledge about HPV vaccine

Overall, 281 (45.3%) female preparatory school students have good knowledge about the HPV vaccine (Fig 2). More than half of respondents 320(51.6%) knew the HPV vaccine prevents HPV infection and more than two third of participants 472(76.1%) knew that the HPV vaccine prevents cervical cancer. More than half of the participants 361(58.2%) knew the HPV vaccine should be given before the first sexual intercourse and 291(46.9%) knew the HPV vaccine can be given to people who have had a sexual history. More than half of the respondents 334

**Table 1. Socio-demographic characteristics of female preparatory school students in Bahir Dar city, Ethiopia, 2021(n = 620).**

| Variables | Category | Frequency | Percent |
|---|---|---|---|
| **Age** | 15–19 | 555 | 89.5% |
| | 20–24 | 65 | 10.5% |
| **Religion** | Orthodox | 482 | 67.7% |
| | Muslim | 79 | 12.7% |
| | Catholic | 32 | 5.2% |
| | Protestant | 27 | 4.4% |
| **Grade level** | Grade 11 | 222 | 35.8% |
| | Grade 12 | 398 | 64.2% |
| **Parents educational status** | Informal education | 84 | 13. 5% |
| | Primary and secondary | 349 | 56.3% |
| | Diploma and above | 187 | 30.2% |
| **Parents occupation** | House wife | 179 | 29% |
| | Employed | 240 | 39% |
| | Merchant | 163 | 26% |
| | Others | 38 | 6% |
| **Average family monthly income** | <4000 | 85 | 13.7% |
| | 4000–7000 | 336 | 54.2% |
| | >7000 | 199 | 32.1% |

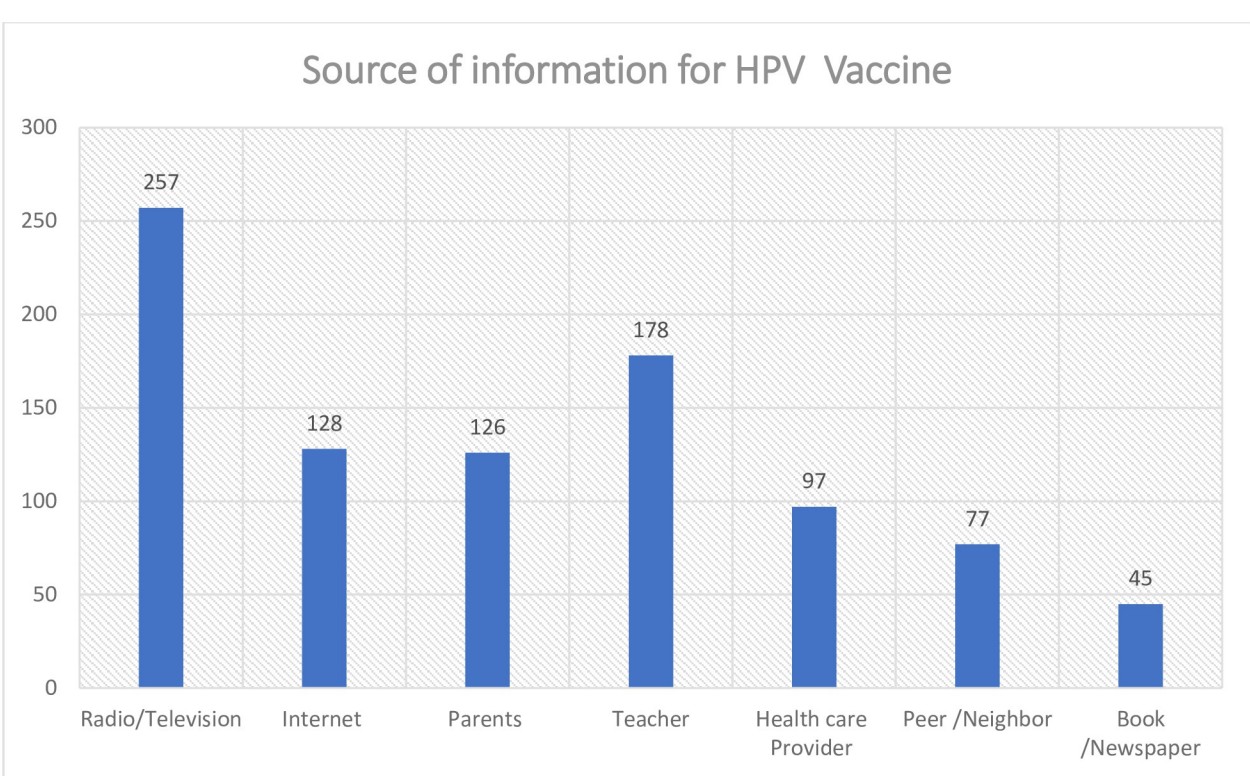

**Fig 1. Source of information about HPV vaccine, Bahir Dar, Ethiopia, 2021.**

(53.9%) knew the recommended age for HPV vaccine vaccination, which is between the ages of 10 and 14 years, but almost two-thirds of participants 399 (64.4%) did not know the recommended dose of HPV vaccine (Table 2).

## Attitude towards HPV vaccine

The overall positive attitude toward the HPV vaccine among respondents was 360 (58%). More than three-fourth of the respondents 478(77%) agree that the HPV vaccine is effective in preventing cervical cancer, and more than two third of respondents 408(65.8%) agree that they will take the vaccine because they believe they are at risk of getting HPV infection, while 392 (63.2%) believe that the vaccine's side effects will not deter them from taking the vaccine. Nearly three-quarters of participants 440(71%) agreed that getting the HPV vaccine before becoming sexually active is preferable, and 472(76%) believe that more information about HPV and its vaccine is needed before getting the vaccine (Table 3).

## Uptake of HPV vaccine

Only 102 (16%) of respondents had received the HPV vaccine, with one-thirds (33%) having completed the vaccine doses (Fig 3). Of the total participants, 518 (84%) were not vaccinated against HPV in the study area. The main reason for not receiving the HPV vaccine was not being informed by the health care providers 79(15.3%), a lack of knowledge about the vaccine's safety 270(52%), don't know where the vaccine is obtained 93(18%), and a lack of parental support 76(14.7%).

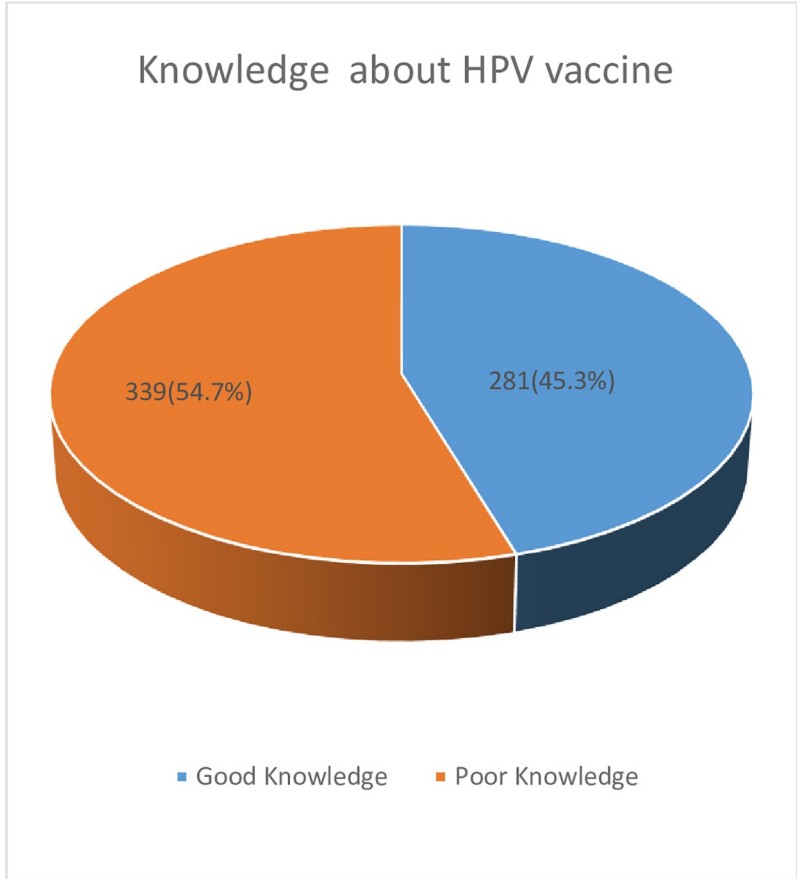

**Fig 2. Knowledge of respondents about HPV vaccine, Bahir Dar, Ethiopia, 2021.**

## Factors associated with knowledge of the human papillomavirus vaccination

Multivariable logistic regression revealed that having a history of sexual contact, hearing about HPV infection, and having a positive attitude toward the HPV vaccine were all significantly associated with knowledge about the HPV vaccine. Students who have had a history of sexual contact increased almost 3 times the odds of knowledge about the HPV vaccine as compared to their counterparts [AOR = 2.8, 95% CI = (1.64–4.76)]. Participants who had heard about HPV infection were 1.6 times more likely to have increased knowledge about the HPV vaccine than those who had not heard about HPV infection [AOR = 1.59. 95% CI = (1.13–2.24)]. Participants with a positive attitude toward the HPV vaccine were 1.46 times more likely to increase their knowledge about the HPV vaccine than those with a negative attitude toward the HPV vaccine. [AOR = 1.46, 95%CI = (1.03–2.08] (Table 4).

## Factors associated with attitudes toward the human papillomavirus vaccination

Participants who discussed reproductive health issues with their partners were nearly three times more likely to favor the HPV vaccine [AOR = 2.56,95% CI = (1.80–3.64)]. Students who were well-informed about the HPV vaccine had nearly four times the positive attitude toward the HPV vaccine [AOR = 3.57, 95% CI = (2.49–5.11)] (Table 5).

**Table 2. Knowledge toward HPV vaccination among female preparatory school students, Bahir Dar, Ethiopia, 2021(n = 620).**

| Knowledge items | Category | Frequency | % |
|---|---|---|---|
| Does human papilloma virus vaccine prevent human papilloma virus infection? | Yes | 320 | 51.6% |
| | No | 300 | 48.4% |
| Does human papilloma virus vaccine prevent cervical cancer? | Yes | 472 | 76.1% |
| | No | 148 | 23.9% |
| Is the human papillomavirus vaccine used to prevent genital warts? | Yes | 320 | 51.6% |
| | No | 300 | 48.4% |
| Should a human papilloma virus vaccine be given prior to the first sexual intercourse? | Yes | 361 | 58.2% |
| | No | 259 | 41.8% |
| Can human papilloma virus vaccine be given to people who have had sex? | Yes | 291 | 46.9% |
| | No | 329 | 53.1% |
| Can human papillomavirus vaccine be given to a woman who is already infected with HPV? | Yes | 285 | 46.0% |
| | No | 335 | 54.0% |
| What age range is recommended for vaccination against human papillomavirus infection? | 10-14years | 334 | 53.9% |
| | No | 286 | 46.1% |
| How many doses are recommended for a human papillomavirus vaccine? | Two does | 221 | 35.6% |
| | No | 399 | 64.4% |
| Did you know the schedule for a human papillomavirus vaccination? | 6 month-1year | 217 | 35.0% |
| | No | 403 | 65.0% |
| Did you know the site of vaccine for HPV vaccination? | Arm | 195 | 31.5% |
| | No | 425 | 68.5% |

## Factors associated with uptake of human papillomavirus vaccination

Participants with good knowledge of the HPV vaccine were 2.36 times more likely to receive the HPV vaccine than those who had poor knowledge [AOR = 2.36, 95% CI = (1.48–3.76)]. Students with a favorable attitude toward the HPV vaccine were nearly three times more likely to the uptake of HPV vaccine [AOR = 2.87, 95% CI = (1.70–4.85)] (Table 6).

## Discussion

The primary goal of this study was to provide insight into the level of HPV vaccine knowledge, attitude, and uptake among school students, as well as to identify potential factors associated with HPV vaccine knowledge, attitude, and uptake.

**Table 3. Attitude toward HPV vaccination among female preparatory school students, Bahir Dar, Ethiopia, 2021(n = 620).**

| Attitude items | Agree | Disagree | Undecided |
|---|---|---|---|
| | n (%) | n (%) | n (%) |
| HPV vaccine is effective in preventing cervical cancer | 478(77) | 75(12.1) | 67(10.8) |
| I will take the vaccine because I feel at risk of getting HPV infection | 408(65.8) | 100(16.1) | 112(18.1) |
| Person who has only one sexual partner can protect from HPV infection | 378(61) | 79(15.7) | 146(32.5) |
| It's not necessary to get the human papilloma virus vaccination | 177(23.8) | 304(49) | 94(15.2) |
| I believe the vaccine's side effects are reasonable and will not prevent me from receiving the vaccine | 392(63.2) | 97(15.7) | 131(21.1) |
| I feel it is better to be vaccinated before becoming sexually active | 440(71) | 59(9.5) | 121(19.5) |
| More information on HPV and its vaccine will be needed before I take the vaccine. | 472(76) | 47(8) | 101(16) |
| Human papilloma virus vaccine may have long negative effect | 294(47.4( | 159(25.7) | 157(26.9) |
| I feel only sexually active ladies should get the vaccine | 238(38.4) | 236(38) | 146(23.6) |
| Do you think your family allow to vaccinate | 254(41) | 202(32.6) | 164(26.4) |
| Education on HPV vaccine should be implemented at school | 469(75.7) | 72(11.6) | 79(12.7) |
| HPV vaccination should be included on the National Program on immunization. | 462(74.5) | 71(11.5) | 87(14) |

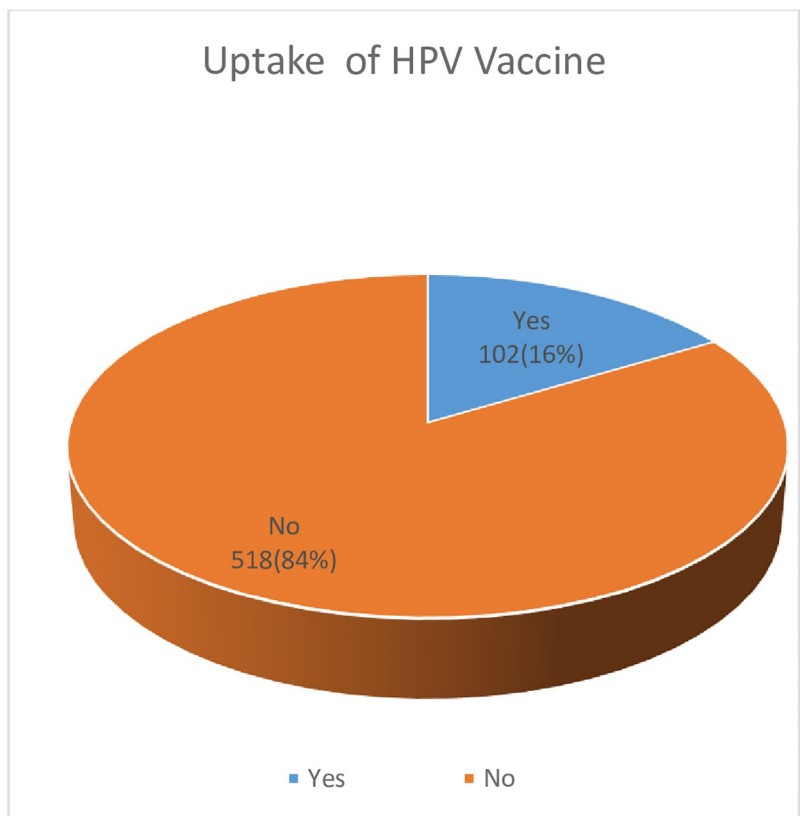

**Fig 3. Uptake of HPV vaccine among female preparatory school students, Bahir Dra, Ethiopia, 2021.**

This study revealed that the knowledge about of HPV vaccine among preparatory female students in the study area was found to be 45.3%. This finding is consistent with the findings of other studies in Debre Markos, Ethiopia(47.6%) [20], Indonesia (44.4%) [21], India (44%) [22]. However, this finding is relatively lower than that of study in Brazil at 51.79% [23], Italy at 56.3% [24], and Malaysia at 50.8% [25]. This study found that 58% of the preparatory female students in the study area had a favorable attitude toward the HPV vaccine. This is consistent

**Table 4. Factors associated with knowledge about HPV vaccine among female preparatory school students, Bahir Dar, Ethiopia, 2021.**

| Variables | Knowledge about HPV vaccination | | COR (95% CI) | AOR (95%CI) |
|---|---|---|---|---|
| | **Good** | **Poor** | | |
| **Had ever sexual contact history** | | | | |
| Yes | 93 | 45 | 3.23(2.17–4.82) | 2.80(1.64–4.76) |
| No | 188 | 284 | 1 | 1 |
| **Heard about HPV infection** | | | | |
| Yes | 160 | 155 | 1.57(1.14–2.16) | 1.59(1.13–2.24) |
| No | 121 | 184 | 1 | 1 |
| **Attitude about HPV vaccine** | | | | |
| Positive attitude | 178 | 182 | 1.49(1.08–2.05) | 1.46(1.03–2.08) |
| Negative attitude | 103 | 157 | 1 | 1 |

**Table 5. Factors associated with attitude about HPV vaccination among female preparatory school students, Bahir Dar, Ethiopia, 2021.**

| Variables | Attitude towards HPV vaccine | | COR (95%CI) | AOR (95%CI) |
|---|---|---|---|---|
| | Positive Attitude | Poor Attitude | | |
| **Discussion of RH issues with family** | | | | |
| Yes | 147 | 78 | 3.22(2.25–4.62) | 2.56(1.80–3.64) |
| No | 213 | 182 | 1 | 1 |
| **Knowledge about HPV Vaccine** | | | | |
| Good knowledge | 210 | 71 | 3.73(2.62–5.26) | 3.57(2.49–5.11) |
| Poor knowledge | 150 | 189 | 1 | 1 |

with a study conducted in Nigeria, which found 61.8% [26]. However, this result is less than those from India (69%) and Benin-City, Nigeria (61.8%) [27, 28].

This study revealed that only 16% of preparatory female students in Bahir Dar town were vaccinated with the HPV vaccine. This is slightly in line with studies done among female college students in mainland China at 11% [29], Mbale District, Uganda at 14% [30], and 16% of the female college students in Kenya had completed the full recommended two-dose schedule [31]. However, this result is significantly lower than the Ugandan study's 42.4% [32], Germany's 53% [33], and the United States' 62.4% [34]. This could be a result of the different sample sizes, study population, and strategies for the healthcare delivery system.

Teachers and the media (radio, television, and the internet) were the main sources of information about HPV vaccination, followed by parents and medical professionals. This is supported by research from Indonesia [35], Nigeria [36], and Turkey [37].

Students with a history of sexual contact, heard about HPV infection, and a positive attitude toward the HPV vaccine had a significant association with HPV vaccine knowledge. Students who discussed reproductive health issues with their partners and those who had good knowledge about HPV infection were significantly associated with positive attitudes toward HPV vaccination. Participants with good knowledge of the HPV vaccine and a favorable attitude toward the HPV vaccine were significantly associated with HPV vaccine uptake.

Students who have a history of sexual contact are more likely to know about the HPV vaccine by 2.8 times. This is consistent with research conducted in Lagos State, Nigeria [38]. The possible reason might be due to students with a history of sexual activity who may fear STIs, HIV, and pregnancy-related issues. They also may be aware that the probability of getting HPV from sexual contact is high. Students ever heard about the HPV vaccine before the study increased 1.59 times the odds of knowledge of the HPV vaccine. This is in line with a study in Brazil [39]. The possible reason might be due to students who have heard about the HPV

**Table 6. Factors associated with uptake of HPV vaccine among female preparatory school students, Bahir Dar, Ethiopia, 2021.**

| Variables | Uptake of HPV vaccine | | COR | AOR |
|---|---|---|---|---|
| | Yes | No | | |
| **Knowledge about HPV vaccination** | | | | |
| Good knowledge | 65 | 216 | 2.46(1.58–3.82) | 2.36(1.48–3.76) |
| Poor knowledge | 37 | 302 | 1 | 1 |
| **Attitude about HPV vaccine** | | | | |
| Positive attitude | 79 | 281 | 3.17(1.76–4.76) | 2.87(1.70–4.85) |
| Poor attitude | 23 | 237 | 1 | 1 |

vaccine may read about it or ask others about its benefits. This increases their understanding of the HPV vaccine's disadvantages and side effects.

Students with a positive attitude toward the HPV vaccine increased their chances of knowing about the HPV vaccine by 1.46 times. This is consistence with a study conducted in Debre Markos town [20]. This is might be due to the fact that individuals who have a positive attitude towards the vaccine may utilize it. Students' discussion of the reproductive issue with their parents increased the odds of attitude towards the HPV vaccine by 2.56 times. Parents who are transparent about sexuality with their young children have more robust communication, which is crucial for preventing risky sexual activities like early sexual initiation, unwanted pregnancy, and other reproductive health issues.

Having awareness about the HPV vaccine increased the attitude toward vaccination of HPV vaccine by 3.57 times. This is consistent with the study done in Debre Markos town, Ethiopia [20]. Having a satisfactory level of knowledge about the HPV vaccine increased their uptake of vaccination by 2.36 times. This is in line with research conducted in different parts of Ethiopia, including Ambo Town, Addis Ababa Town, and Gondar Town. [40–42].

HPV vaccination is influenced by practice factors, such as adequate health information systems that include provider or patient prompts for vaccine administration [43]. Supporting evidence indicates that adolescents are more likely to receive the HPV vaccine before they begin having sex when they are adequately informed about the vaccines and realize that the vaccine is safe, HPV-related diseases are serious, and adolescents are susceptible. [44]. Knowledge of HPV is important in how people think about their sexuality, protection, and prevention [45]. Having a positive attitude towards the HPV vaccine increased the uptake of the HPV vaccine by 2.87 times. This is a consistent study conducted in Benin City, Nigeria [27], and Uganda [46]. This may be explained by the fact that the majority of motivating/initiative factors in adolescents' practice are generated by their positive attitude.

## Conclusion

The respondent's knowledge of HPV and attitude toward the HPV vaccine was generally poor, and the uptake of the HPV vaccine was very low. Knowledge of the HPV vaccine was significantly associated with a history of sexual contact, hearing about HPV infection, and having a favorable attitude toward the HPV vaccine. Discussion of reproductive health issues with their families, and knowledge of the HPV vaccine, were found to be associated with a favorable attitude toward the HPV vaccine. Good knowledge and a favorable attitude toward the HPV vaccine were strongly associated with HPV vaccine uptake. The main identified barriers to HPV vaccine uptake were not being informed by health care providers, a lack of knowledge about the vaccine's safety, not knowing where the vaccine is obtained, and a lack of parental support.

## Recommendations

We recommend the Ethiopian Ministry of Health conduct mass HPV vaccinations and community sensitizations to foster positive attitudes and ensure consistent vaccine availability at vaccination sites. For ease of access, the government should make an effort to add the HPV vaccine to the National Program on Immunization (NPI) schedule. All stakeholders should collaborate to improve female students' knowledge and attitudes toward HPV vaccination.

## Supporting information

**S1 File. English version questionnaire.**
(DOCX)

**S2 File. SPSS data.**
(SAV)

## Acknowledgments

We would like to acknowledge Bahirdar University College of Medicine and Health Sciences for their Ethical approval. Our deepest gratitude goes to our participants for their cooperation during the study as well as their permission to publish this article.

## Author Contributions

**Conceptualization:** Etenesh Adela Lakneh.

**Data curation:** Etenesh Adela Lakneh, Eleni Admassu Mersha, Melash Belachew Asresie.

**Formal analysis:** Etenesh Adela Lakneh, Melash Belachew Asresie, Habtamu Gebrehana Belay.

**Methodology:** Etenesh Adela Lakneh, Eleni Admassu Mersha, Melash Belachew Asresie, Habtamu Gebrehana Belay.

**Software:** Eleni Admassu Mersha, Melash Belachew Asresie, Habtamu Gebrehana Belay.

**Supervision:** Etenesh Adela Lakneh.

**Validation:** Eleni Admassu Mersha, Melash Belachew Asresie, Habtamu Gebrehana Belay.

**Visualization:** Eleni Admassu Mersha, Melash Belachew Asresie, Habtamu Gebrehana Belay.

**Writing – original draft:** Etenesh Adela Lakneh, Habtamu Gebrehana Belay.

**Writing – review & editing:** Etenesh Adela Lakneh, Eleni Admassu Mersha, Melash Belachew Asresie, Habtamu Gebrehana Belay.

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
