## [Decision Letter · Decision Letter 0]

20 Sep 2022

PONE-D-22-21861Knowledge, Attitude, and Uptake of Human Papilloma Virus Vaccine and Associated Factors Among Preparatory School Female Students in Bahir Dar City, Northwest, EthiopiaPLOS ONE

Dear Dr. Belay,

Thank you for submitting your manuscript to PLOS ONE. After careful consideration, we feel that it has merit but does not fully meet PLOS ONE’s publication criteria as it currently stands. Therefore, we invite you to submit a revised version of the manuscript that addresses the points raised during the review process.

We look forward to receiving your revised manuscript.

Kind regards,

Kehinde Sharafadeen Okunade

Academic Editor

PLOS ONE

Journal Requirements:

"no specific funding for this work"

"authors have declared that no competing interests exist."

Reviewers' comments:

Reviewer's Responses to Questions

**Comments to the Author**

1. Is the manuscript technically sound, and do the data support the conclusions?

Reviewer #1: Yes

Reviewer #2: Yes

2. Has the statistical analysis been performed appropriately and rigorously? 

Reviewer #1: No

Reviewer #2: Yes

3. Have the authors made all data underlying the findings in their manuscript fully available?

Reviewer #1: Yes

Reviewer #2: Yes

4. Is the manuscript presented in an intelligible fashion and written in standard English?

Reviewer #1: No

Reviewer #2: Yes

5. Review Comments to the Author

Reviewer #1: The author needs to review the spelling and tenses in the manuscript. Also, in the result section, the author needs to rewrite it for clarity. I have highlighted and written comments on areas that need attention in the manuscript.

Reviewer #2: 1. bearing in mind the several problems have been identified with the HL test. For example, it doesn’t take overfitting into account and tends to have low power, What informed the choice of Hosmer and Lemeshow goodness of fit in your methodology

2. Is it descriptive or analytical cross sectional study

3. Its an emerging fact that a single-dose Human Papillomavirus (HPV) vaccine delivers solid protection against HPV, the virus that causes cervical cancer, that is comparable to 2-dose schedules, NO MENTION OF IT IN YOUR MANUSCRIPT

6. PLOS authors have the option to publish the peer review history of their article (what does this mean?). If published, this will include your full peer review and any attached files.

Reviewer #1: No

Reviewer #2: **Yes: **Aloy Okechukwu Ugwu

---

## [Editor Report · Decision Letter 1]

7 Oct 2022

Knowledge, Attitude, and Uptake of Human Papilloma Virus Vaccine and Associated Factors Among Female Preparatory School Students in Bahir Dar City, Amhara Region, Ethiopia

PONE-D-22-21861R1

Dear Dr. Belay,

We’re pleased to inform you that your manuscript has been judged scientifically suitable for publication and will be formally accepted for publication once it meets all outstanding technical requirements.

Kind regards,

Kehinde Sharafadeen Okunade

Academic Editor

PLOS ONE
---

## [Editor Report · Acceptance letter]

8 Nov 2022

PONE-D-22-21861R1 

Knowledge, Attitude, and Uptake of Human Papilloma Virus Vaccine and Associated Factors Among Female Preparatory School Students in Bahir Dar City, Amhara Region, Ethiopia 

Dear Dr. Belay:

I'm pleased to inform you that your manuscript has been deemed suitable for publication in PLOS ONE. Congratulations! Your manuscript is now with our production department. 

Kind regards, 

on behalf of

Dr. Kehinde Sharafadeen Okunade 

Academic Editor

PLOS ONE